# Viral Hepatitis B and Its Implications for Public Health in DR Congo: A Systematic Review

**DOI:** 10.3390/v17010009

**Published:** 2024-12-25

**Authors:** Marcellin Mengouo Nimpa, Hermès Karemere, Christian Ngandu, Franck-Fortune Mboussou, M. Carolina Danovaro-Holliday, Dalau Nkamba, André Bita Fouda, Bienvenu Nguejio, Sifa Kakozi, Aimé Mwana-Wabene Cikomola, Jean-Crispin Mukendi, Dieudonné Mwamba, Moise Désiré Yapi, Richard Bahizire Riziki, Cedric Mwanga, John Otomba, Jean Baptiste Nikiema, Boureima Hama Sambo, Daniel Katuashi Ishoso

**Affiliations:** 1World Health Organization (WHO) Country Office, Kinshasa 01206, Democratic Republic of the Congo; nimpamengouom@who.int (M.M.N.); nguejiob@who.int (B.N.); sifakakozi@gmail.com (S.K.); yapimo@who.int (M.D.Y.); bahizirer@who.int (R.B.R.); mwangac@who.int (C.M.); otombatondaepengej@who.int (J.O.); nikiemaje@who.int (J.B.N.); sambob@who.int (B.H.S.); 2École Régionale de Santé Publique, Université Catholique de Bukavu, Bukavu 11103, Democratic Republic of the Congo; hkaremere@gmail.com; 3National Institute of Public Health, Kinshasa 01206, Democratic Republic of the Congo; nganduchristian@ymail.com (C.N.); dk.mwamba@umontreal.ca (D.M.); 4World Health Organization African Regional Office, Brazzaville P.O. Box 06, Congo; mboussouf@who.int (F.-F.M.); abita@who.int (A.B.F.); 5Immunization, Analytics and Insights (IAI), Department of Immunization, Vaccines and Biologicals (IVB), World Health Organization (WHO), 1211 Geneva, Switzerland; danovaroc@who.int; 6Kinshasa School of Public Health, University of Kinshasa, Kinshasa 01206, Democratic Republic of the Congo; dalau.nkamba@unikin.ac.cd; 7Expanded Program of Immunization, Kinshasa 01206, Democratic Republic of the Congo; aimcik@yahoo.fr (A.M.-W.C.); mukendijean2@gmail.com (J.-C.M.); 8Higher Institute of Medical Techniques of Nyangezi, Public Health Section, Sud-Kivu 11213, Democratic Republic of the Congo

**Keywords:** Democratic Republic of the Congo, viral hepatitis B, public health

## Abstract

**Background:** The prevalence of hepatitis B virus infection remains high in the Democratic Republic of Congo (DRC), constituting a public health problem in view of the fatal complications it causes, notably cirrhosis and hepatocellular carcinoma. The aim of this study was to provide an overview of the situation of viral hepatitis B in the DRC and in particular its implications for public health. **Methods:** A systematic review was conducted according to Preferred Reporting Items for Systematic Reviews and Meta-Analyses (PRISMA) group guidelines. Google Scholar, PubMed, and ResearchGate were used as databases. The review essentially analyzed the viral hepatitis B (HBV) vaccination status of study subjects, diagnostic means, HBV genotypes in DR Congo, seroprevalence of HBV infection, subjects’ level of knowledge and perceptions of HBV, co-infection/comorbidity with HBV infection, factors associated with HBV infection and public health issues raised by HBV infection. **Results:** The vast majority of studies (69%) were carried out to determine the seroprevalence of HBV infection. The Determine rapid test was the most widely used test (10 studies), sometimes combined with ELISA (3 studies) and polymerase chain reaction (PCR) (1 study, for genotyping). Some of the public health issues raised by hepatitis B virus infection were identified in the course of the included studies, in relation to co-infection, comorbidity, associated factors, and individuals’ level of knowledge and perceptions of HBV. Certain factors were identified as being closely associated with HBV, notably healthcare professions (e.g., doctor, laboratory technician) and having several sexual partners. In terms of perception of HBV, the hepatitis B virus is recognized as dangerous, and the majority of people questioned in the various studies were aware that vaccination remains the most effective means of prevention. **Conclusions:** Hepatitis B is a highly contagious infectious disease present in the DRC, with a higher prevalence among healthcare professionals, sex workers, patients with certain diseases including HIV, and people with a history of blood transfusion. The surveillance system within the national blood transfusion program needs to be strengthened. Raising public awareness of the seriousness of viral hepatitis B, offering vaccination to at-risk populations, and systematically screening pregnant women and blood donors for HBV infection could help reduce the prevalence of viral hepatitis B.

## 1. Introduction

Hepatitis B virus (HBV) infection remains one of the world’s major public health problems. The World Health Organization (WHO) estimates that 296 million people will be living with chronic hepatitis B in 2019, with 1.5 million new infections every year. In 2019, hepatitis B caused around 820,000 deaths, mainly from cirrhosis or hepatocellular carcinoma [1]. The chronic course of HBV infection thus increases the risk of progressive liver disease and premature death from complications such as cirrhosis, liver failure, and hepatocellular carcinoma [2]. The main transmission routes include blood, mother-to-child transmission, and sexual transmission, which is more prevalent in unvaccinated individuals with multiple sexual partners [3]. A safe and effective vaccine exists, capable of reducing the incidence of acute hepatitis B and then lowering the risk of chronic infections and their consequences in high- and low-endemic regions [4].

HBV is a DNA virus of the *Hepadnaviridae* family and the Orthohepadnavirus genus. It is subdivided into 10 genotypes (A to J) and some 48 subgenotypes, with different biological properties and geographical distributions [5]. It has been reported that HBV genotype may influence disease progression, response to antiviral treatment, and possibly vaccination against the virus [6,7].

Most newly infected individuals are asymptomatic. The main symptoms in people with acute disease are jaundice, dark urine, extreme fatigue, and digestive symptoms (mainly nausea, vomiting, and abdominal pain). Acute hepatitis can lead to life-threatening acute liver failure.

Co-infections with HIV, HBV, and/or HCV are currently a major public health problem, especially in view of their interaction with the natural history of chronic hepatitis, which they aggravate (faster progression to cirrhosis and hepatocarcinoma), leading to high mortality and difficult therapeutic problems [8].

In developing countries, high-risk groups for infection include people requiring blood products, newborn babies, pregnant women, and health workers [9]. For HBV diagnosis, laboratory tests focus not only on the detection of hepatitis B surface antigen (HBsAg) but also on other serology markers such as anti-HBc and HBeAg. HBV DNA is an important biomarker to predict viral load and evaluate treatment efficacy. HBsAg testing tends to give false positives or false negatives (due to immune escape mutations) and tends to be affected by core promoter mutations [7,10].

Despite these diagnostic efforts, significant knowledge gaps remain regarding the epidemiology of hepatitis B in the Democratic Republic of Congo (DRC). Challenges related to data collection, vaccination barriers, and diagnostic limitations exemplify these knowledge gaps on HBV in the DRC. The DRC is classified as a country with an intermediate to high prevalence of HBV infection, with an estimated seroprevalence of about 5% of the population [3]. However, there is no reliable national survey to determine HBV seroprevalence in the DRC or to assess its public health impact. Previous studies on HBV in the DRC have mainly focused on national or provincial HBV seroprevalence, HBV genotypes, or levels of knowledge and perceptions about HBV. Too few studies have addressed the vaccination status of the population against HBV infection, the means of diagnosing HBV infection, the factors associated with HBV infection, or the public health impacts caused by HBV infection.

The aim of this review is to synthesize knowledge about hepatitis B in the DRC and to fill the gaps in knowledge about the public health challenges posed by HBV infection in the country.

## 2. Materials and Methods

This systematic review followed the guidelines of the PRISMA group in line with systematic reviews and meta-analysis reporting [11]. A research librarian was consulted in developing the search strategy and the selection criteria. The systematic review protocol was registered in the International Prospective Registry of Systematic Reviews (PROSPERO), with registration number CRD42024614692 (https://www.crd.york.ac.uk/prospero/#recordDetails, accessed on 20 December 2024).

### 2.1. Bibliographic Research Approach

A team of two researchers carried out this systematic review. It included all publications from 2007 to 2023 reporting information on hepatitis B in the DRC. The literature search initially covered a wider range of topics, selecting publications from prospective, retrospective, case-control, cross-sectional, and descriptive studies focusing on viral hepatitis B in the DRC.

The search based on published literature was conducted according to the following keywords, in French and English: hepatitis B, cirrhosis, Democratic Republic of Congo, DRC, prevalence, impact, incidence, knowledge, perceptions, hepatitis B surface antigen, HBsAg prevalence, and hepatitis B sero-epidemiology. A previous systematic review on the prevalence of viral hepatitis B in the DRC [3] was analyzed for comparison.

A three-stage literature search strategy was deployed [12]. An initial limited search on PubMed, Google Scholar, and ResearchGate was undertaken, followed by an analysis of the words contained in the title and finally the reading of abstracts in order to select the articles to be included in the study.

### 2.2. Article Selection

A previously developed article selection grid was applied to each article. This grid comprised a series of criteria including the name of the first author and the year of publication of the article, the title of the article, the DRC province(s) included, the type of study, criteria of rigorous methodology, the context of the study, the purpose of the study, the results of the study and the reviewer’s commentary. The selection of articles was carried out by two researchers, and articles were retained after analysis of each abstract, taking into account only their year of publication, their relevance, and their contribution to the objectives of the study. Discordant choices were discussed between the two researchers to decide whether or not to include them.

The PICOS model (population, intervention, comparison, outcome, study design) was used to guide article selection [13]. Inclusion and exclusion criteria, based on the PICOS model, included studies involving patients with hepatitis B virus (HBV) in the Democratic Republic of Congo (DRC) for the population; studies addressing laboratory examinations and treatments for HBV in the DRC for the intervention; all studies addressing HBV whether comparative or not, for the comparison; and all studies conducted in the DRC on HBV, regardless of their outcomes, for the outcome criteria. For the study design, we included all studies conducted in the DRC from 2007 onwards, a deliberate choice to align with recent recommendations. All other studies not meeting these criteria were excluded from selection.

### 2.3. Data Extraction

A previously developed data extraction grid was applied to each article selected for the study, allowing standardization of the data extraction methodology. The extraction tool included the following variables: search results, study characteristics, diagnostics and genotypes, seroprevalence, and public health challenges.

### 2.4. Data Analysis and Synthesis

The extracted data were entered into Microsoft Excel. In view of the research question aimed at providing information on viral hepatitis B in DRC and its implications for public health, a qualitative, theme-based analysis was preferred. The main themes analyzed concerned the location (DRC towns and provinces) involved in the studies on HBV infection, the main purpose of the studies, the study subjects, the HBV vaccination status of the study subjects, diagnostic means, HBV genotypes, seroprevalence of HBV infection, subjects’ level of knowledge and perceptions of HBV, co-infection/comorbidity with HBV infection, factors associated with HBV infection, and public health issues posed by HBV infection.

### 2.5. Ethical Considerations

The present study did not require ethical approval. It was based on data extracted from published, publicly available articles. No personal or confidential data were involved.

## 3. Results

### 3.1. Results of the Bibliographic Search for the Review

The search identified a total of 148 potentially relevant publications. After excluding 77 duplicate publications (same publication in English and French, present in two or more databases), 71 publications were analyzed. Of these, 21 were excluded based on their title and/or abstract as unrelated to the research question. Next, 50 articles were analyzed in their entirety, and 20 were excluded on the basis of their relevance (study not related to DRC or subject of the study not related to the research question). Thus, 30 publications were selected and included in the present review (Figure 1 and Table 1).

### 3.2. Study Characteristics

Of the 30 studies included, 6 were carried out in South Kivu [8,14,15,16,17,18], 5 in Kinshasa [19,20,21,22,23], 5 in Haut-Katanga [6,24,25,26,27], 3 in Tshopo province [28,29,30], 3 in Kongo-Central [31,32,33], 1 in North Kivu [34], 1 in Maniema [35], 1 in Lualaba [36], 1 in Sud-Ubangi [37], and 4 concerned the whole of DR Congo [3,38,39,40]. Some of these studies concerned either all the provinces or certain towns as illustrated in Table 1.

**Table 1 viruses-17-00009-t001:** Publications by province, research subject, and study population.

Author and Year of Publication		Setting	Object of the Study	Population	Sample Size	Co-Infection or Comorbidity Explored
1	Van Aalsburg, 2011 [19]	Kinshasa	Hepatitis B associated with urticaria and periorbital edema	A male patient who visited Kinshasa	1	Urticaria and periorbital edema
2	Kabamba, 2021 [6]	Lubumbashi, Haut-Katanga	Hepatitis B virus (HBV) strains and HBV prevalence	Blood donors	1512	
3	Abdala, 2016 [35]	Kindu, Maniema	HBV seroprevalence	Blood donors	435	HIV
4	Situakibanza, 2017 [32]	Matadi, Kongo-Central	HBV seroprevalence	Blood donors	16	HIV, HVC
5	Lungosi, 2019 [33]	Kisantu, Kongo-Central	HBV seroprevalence	Health workers	97	
6	Miyanga, 2023 [14]	Sud-Kivu	HBV Seroprevalence	Children with HIV vs. those without HIV	594	HIV
7	Chirambiza, 2018 [15]	Bukavu, Sud-Kivu	Blood transfusion profile	Blood donors	357	Syphilis, HIV et HVC
8	Kakisingi, 2016 [26]	Lubumbashi, Haut-Katanga	Blood donor profile	Blood donors	599	
9	Kombi, 2018 [30]	Kisangani, Tshopo	HBV seroprevalence	Diabetics and blood donors	149 (volunteer donors) and 138 diabetics	
10	Kabinda, 2019 [25]	Lubumbashi, Haut-Katanga	Residual risk of HBV transmission during blood transfusion	Blood donors	3149	
11	Mbendi, 2018 [20]	Kinshasa	Cirrhosis (epidemiological, clinical, and evolution aspects)	Cirrhosis patients	1056	
12	Kabinda, 2010 [8]	Bukavu, Sud-Kivu	Prevalence of HIV co-infection with HBV and HCV	People with HIV vs. controls	209 (HIV+) and 211 (HIV−)	
13	Kabamba, 2020 [24]	Lubumbashi, Haut-Katanga	Prevalence of hepatitis D in Blood donors who tested + for HBV	Blood donors who tested positive for HBV	200	(test for HVD)
14	Kabinda, 2014 [16]	Bukavu, Sud-Kivu	Hepatitis B and C seroprevalence	Paid blood donors	1079	HBV-HCV 2.2%
15	Ngalula, 2018 [27]	Lubumbashi, Haut-Katanga	Seroprevalence and HBV risk factors in Lubumbashi	Pregnant women	269	
16	Mpody, 2019 [21]	Kinshasa	Seroprevalence and HBV risk factors in Kinshasa	Pregnant women	1377	HIV
17	Shindano, 2018 [3]	DR Congo	HBV seroprevalence	General population (systematic review)	154,926	
18	Kalimira, 2018 [41]	Minova, Sud-Kivu	General knowledge about HBV infection	Volunteer blood donors	111	
19	Bassandja, 2018 [28]	Isangi, Tshopo	Seroprevalence of viral hepatitis B and C and HIV	Blood donors	814	
20	Mosiba, 2022 [37]	Zongo, Sud-Ubangi	Prevalence of Viral Hepatitis B and HIV/AIDS	Blood donors	100 donors (70 men and 30 women)	HIV
21	Makiala-Mandanda, 2017 [39]	DR Congo	Causality between viral hepatitis A, B, C, D, and E and yellow fever	People with yellow fever	498	
22	Thahir, 2022 [22]	Kinshasa	Pregnant women’s HBV knowledge and perceptions	Pregnant women	220	
23	Sikakulya, 2022 [34]	Butembo, Nord-Kivu	HBV seroprevalence and vaccination status	Health workers	373	
24	Batina, 2007 [29]	Kisangani, Tshopo	HBV seroprevalence	Voluntary, paid, and family donors	3390	
25	Kabinda, 2015 [17]	Sud-Kivu	HBV prevalence and risk factors	Children 6 to 59 months of age	781	
26	Muanza, 2021 [23]	Kinshasa	HBV seroprevalence	Women who have recently given birth and newborns	116 women who have given birth and 118 newborns	
27	Di Masuangi, 2021 [31]	Kimpese, Kongo-Central	Knowledge, attitudes, and practices in relation to HBV	Health workers	91	
28	Gasim, 2015 [38]	DR Congo	Co-infection schistosomiasis and HBV	Patients with chronic hepatitis		Schistosomiasis
29	Mulubwa, 2018 [36]	Kolwezi, Lualaba	HBV seroprevalence	Blood donors	4018	
30	Thompson, 2019 [40]	DR Congo	HBV prevalence	General population (secondary data)	27,483 adults (ages 15–59 years) and 9369 children (ages 0–5 years)	

The majority of studies (69%) focused on determining the seroprevalence of HBV infection across various populations, including children, mother-child pairs, blood donors (volunteer, family, or paid), and individuals with co-infections or comorbidities such as HCV, HVD, HIV, schistosomiasis, urticaria, or diabetes. Among these, one study employed a chi-square test to evaluate vaccination coverage among healthcare workers [33], and another reported prevalence with 95% confidence intervals for HBV in yellow fever cases [39].

Five studies examined HBV associations with specific conditions, including urticaria and periorbital edema [19], cirrhosis [20], yellow fever [39], co-infections with schistosomiasis and HCV [38], and residual HBV transmission risks in blood donors [25]. Most were descriptive studies, with limited statistical analyses, highlighting gaps in methodological rigor.

### 3.3. HBV Diagnostics and Genotypes in the DRC

Screening for viral hepatitis B was carried out by determining the serological profile of the hepatitis B virus, comprising three tests: HBsAg (surface antigen of the hepatitis B virus), HBsAb or anti-HBs (surface antibody of the hepatitis B virus), and HBcAb or anti-HBc (nucleocapsid antibody of the hepatitis B virus). A “positive” or “reactive” HBsAg result confirms acute or chronic infection with the hepatitis B virus. A “positive” or “reactive” HBsAb (or anti-HBs) result indicates a good response to the hepatitis B vaccine or remission of acute hepatitis B infection. Finally, a “positive” or “reactive” HBcAb (or anti-HBc) test result indicates a current or past infection.

In 10 studies, the Determine rapid test was used on its own; in three studies, it was combined with Elisa (enzyme-linked immunosorbent assay), and in one study with PCR (polymerase chain reaction) to identify genotypes. Elisa was used in the only test used in four studies, and PCR alone in one. HBV DNA detection is of interest only in chronic HBsAg carriers, since the diagnosis of acute hepatitis B is based exclusively on serological tests.

HBV genotypes E, A, and D have been the most frequently encountered in the DRC. In a study carried out in Lubumbashi [42], genotypes E and A were present in 53.2% of donors, while genotype A was present in 46.8% of blood donors. In another study, the HBV genotypes present were A, E, and D in patients with yellow fever [39]. A final study identified E and A genotypes in the general DRC population from secondary data [40].

### 3.4. Seroprevalence

Seroprevalence varied depending on the study and the underlying pathology. It ranged from 2% to 8.2% among blood donors, from 0.7% to 8.7% among children with or without HIV, from 4.7% to 6.7% among pregnant women, from 8% to 22.5% among sufferers with co-infection including yellow fever and other viral hepatitis or comorbidity, and an average of 8.2% among healthcare professionals in a study conducted in Kongo-Central [33]. Seroprevalence has not been determined in the general population of the DRC. In some settings, seroprevalence is higher in patients with HIV (8.7%) than in those without (0.7%) [14]. It is similar in diabetic volunteer donors (3.5%) than in nondiabetic volunteer donors (3.4%) [30]. In patients with hepatitis D, 2% developed hepatitis B [6].

### 3.5. Public Health Challenges Posed by HBV Infection

A number of public health issues raised by hepatitis B virus infection have been identified in reviewing the studies. They relate to co-infection, comorbidity, associated factors, and the level of knowledge and perceptions of individuals about HBV.

HBV infection has been associated with urticaria and periorbital edema [19], schistosomiasis [38], and in several other studies with HIV, syphilis, and other hepatic viral infections including A, C, and D. In a study conducted in Kinshasa hospitals [20], 11.3% of cirrhosis cases were caused by HBV. Certain factors are identified as being closely associated with HBV, including healthcare professions (doctor, laboratory technician, etc.) and having multiple sexual partners [33]. In a study conducted in South Kivu, [16], HBV seroprevalence was predominant in healthcare professionals (7.1%), in the under-30 age group (5.0%), in first-time donors (5.1%), and in male subjects (5.1%). This prevalence was statistically significant according to sex (*p* = 0.01) and place of origin (*p* = 0.002). In the same study, there was a strong association between rural areas and hepatitis B: odds ratio (OR) 3.1 (1.4–6.5) and hepatitis C: OR 2.9 (1.3–6.5). The HBV risk donor profile used in the logistic model was a married, rural, male blood donor under 30 years of age.

In another study conducted in Lubumbashi [27], 1.48% (95% confidence interval (CI): 0.41–3.76%) of blood donors were HIV-positive. In the same study, the highest prevalence of hepatitis B was observed in the 31–40 age group (10.53%), among single people (50%), women with gainful occupations (25%), diabetes mellitus (14.29%), and a history of surgery (14.29%). Among pregnant women in Kinshasa, Mpody et al. note an overall high prevalence of HBsAg among those infected with HIV [21].

In addition, the migratory movements imposed by society, the proximity of the border, and commercial exchanges are mentioned as causes of an increase in people living with HIV and hepatitis B in a study carried out in Zongo [37]. Finally, multiparity, jaundice, and blood transfusion were determining factors in the carriage of HBsAg by women giving birth [23,43].

Concerning the perception of HBV, Thahir et al.’s study conducted in Kinshasa [22] notes that 71% of women perceive HBV as a very serious disease while 20% know nothing about HBV. Volunteer blood donors had very low knowledge of HBV in Minova, South Kivu [41]. According to the D-Masuangi study [31] conducted in Kimpese, most caregivers (94.5%) recognized the B virus as the pathogenic agent of HBV, and 89% were aware that vaccination remained the most effective means of prevention.

## 4. Discussion

The aim of this study was to synthesize knowledge about hepatitis B in the DRC and its implications for public health. In the following paragraphs, we discuss methodological limitations, the characteristics of the selected studies, means of diagnosis, HBV seroprevalence, and public health implications.

### 4.1. Methodological Limitations

Unlike a narrative review, a systematic review requires substantial time, the use of multiple databases, and ideally at least three independent reviewers to select and assess the publications to be included [11,44]. In our study, the limited number of databases consulted and the absence of a third reviewer to resolve discrepancies in article selection represent significant limitations. This could introduce a risk of bias or lead to the exclusion of relevant studies, potentially impacting our findings. However, a consensus meeting between the two independent reviewers was conducted to address these discrepancies as effectively as possible.

### 4.2. Characteristics of the Selected Studies

The multiplicity of studies conducted highlights the existence of viral hepatitis B in the DRC. This is alongside other conditions, including vaccine-preventable diseases such as measles, as well as priority endemics, malaria, HIV-AIDS infection, tuberculosis, and two epidemic emerging diseases including Ebola virus disease and coronavirus 2019 [45]. The studies in some cases concern the whole country while in others are predominantly from certain provinces of the country with schools of public health (Bukavu, Kinshasa, Kisangani, and Lubumbashi). Hepatitis B, whether or not associated with other pathologies, is analyzed with a view to determining seroprevalence (often among blood donors, pregnant women, or children); describing epidemiological and clinical aspects; or identifying individuals’ level of HBV knowledge. The present review did not identify any studies on HBV vaccination in the DRC, an important public health intervention aimed at preventing the two major complications of HBV [46], namely cirrhosis and hepatocellular carcinoma. Protection against the highly contagious hepatitis B virus is an international public health issue [47]. The studies identified also fail to address aspects of patient management and clinical course, despite the fact that 5% of infected adults develop chronic hepatitis [48].

### 4.3. Diagnostic Methods

The studies reviewed mainly used rapid screening tests, and very rarely PCR or sequencing. In fact, virological tools useful for the diagnosis, monitoring, and therapeutic management of viral hepatitis linked to HBV are both serological and molecular. Alongside conventional tests for the detection of viral antigens and antibodies directed against them, real-time PCR techniques enable more sensitive and precise quantification of viral DNA, replacing the techniques previously used in most virology laboratories. New markers, such as HBV genotype or the pattern of amino acid substitutions associated with HBV resistance to nucleos(t)idic analogs, can also be characterized using sequencing- or reverse hybridization-based techniques [49].

### 4.4. Seroprevalence

Seroprevalence varies according to the studies analyzed and the population groups considered. In the DRC, the estimated prevalence of HBsAg is 4.9% (95% CI 4.2–5.0), reaching 5.0% (95% CI 4.9–5.1) among blood donors and 5.0% (95% CI 3.0–5.9) among pregnant women [3]. These rates are comparable to the intermediate HBV prevalence levels observed in sub-Saharan Africa, where prevalence typically ranges from 5% to 7% [50]. This situation highlights the need for ongoing monitoring and strengthened vaccination efforts within the national health strategy. Determining seroprevalence in the general population is essential, given the elevated prevalence observed in specific groups and the serious complications associated with this viral liver infection, which place a significant burden on public health.

### 4.5. Public Health Issues and Interventions

One of the main public health challenges in addressing HBV infection in the DRC lies in scaling up vaccination efforts. Although vaccination against hepatitis B has proven effective—with an 85% seroprotection rate among infants born to vaccinated mothers [51]—the implementation of universal early vaccination programs is hindered by logistical and economic barriers. While these programs have reduced the incidence of chronic HBV infection and consequently lowered rates of hepatocellular carcinoma in high-prevalence areas [45], their reach remains limited. To maximize impact, vaccination efforts must target not only infants but also high-risk groups such as healthcare workers, sex workers, and individuals requiring blood transfusions [52].

Moreover, diagnostic limitations pose additional obstacles to effective HBV management. Rapid diagnostic tests (RDTs), primarily used in the DRC, are convenient but have limitations, including lower sensitivity and specificity compared to polymerase chain reaction (PCR) testing and sequencing [53]. The limited use of PCR and sequencing restricts the ability to accurately detect chronic infections, hindering effective case management and public health surveillance. Expanding access to PCR and advanced diagnostic tools could improve detection rates and facilitate better management of chronic cases.

Addressing these challenges requires a multifaceted public health approach, including expanding mobile vaccination clinics, increasing awareness through educational campaigns, and potentially leveraging international aid to support these efforts. Additionally, establishing robust border surveillance systems and improving the capacity for accurate and timely diagnosis could further enhance HBV control and prevention strategies in the DRC [54].

## 5. Conclusions

Hepatitis B is a persistent public health challenge in the DRC, with a notable prevalence among high-risk groups, including healthcare workers, sex workers, individuals with HIV, and those with a history of blood transfusion. Current diagnostic practices, often limited to rapid tests, lack the sensitivity and specificity of more advanced methods such as PCR, which hampers early detection and effective case management. The seroprevalence of HBV varies widely across regions and population groups in the DRC, underscoring the need for regional comparisons, particularly with other Central and sub-Saharan African countries, to contextualize findings and enhance intervention strategies.

This review highlights significant gaps in vaccination coverage and management of HBV cases in the DRC, which, coupled with limited public knowledge and awareness, contribute to the virus’s persistence. Utilizing frameworks such as the WHO’s Global Health Sector Strategy on Viral Hepatitis can provide a structured approach to addressing these gaps. Recommended actions include enhancing the national blood transfusion program’s surveillance system, conducting systematic HBV screening among HIV-positive individuals and pregnant women, and offering targeted vaccinations for at-risk populations. Additionally, overcoming barriers such as inadequate healthcare infrastructure, funding limitations, and political instability will be crucial to strengthening the overall response to hepatitis B. Ultimately, increased public awareness campaigns, improved diagnostic capacity, and expanded vaccination efforts are essential for reducing HBV transmission and its associated health risks in the DRC.

## Figures and Tables

**Figure 1 viruses-17-00009-f001:**
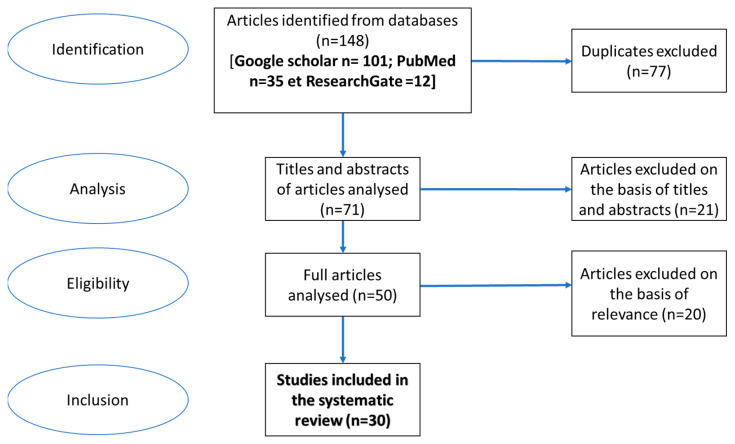
Process for identifying and including articles in the review.

## Data Availability

The data presented in this study are available on request from the WHO-DRC office at the email address “nimpamengouom@who.int”.

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
