# Peer review of "Viral Hepatitis B and Its Implications for Public Health in DR Congo: A Systematic Review"

_viruses, 2024, doi:10.3390/v17010009_

Round 1
Reviewer 1 Report
Comments and Suggestions for Authors
See attached file

Reviewer 2 Report
Comments and Suggestions for Authors
The authors performed a systematic review to collect all relevant data regarding HBV in the Democratic Republic of Congo. Going through the introduction I felt the importance of the topic and the reason for doing it, but the importance was lost through results and discussion. The paper has a quality that can be approved.
Introduction
It is well-written and logic, and subtopics follow continuously. Please add information regarding the development level of DRC – lines 73-75.
Methodology
1. Did you register your systematic review at PROSPERO?
2. Why did you choose publications from 2007? I strongly recommend considering all publications without date or year restrictions.
3. Please report complete search strategies for each of the three electronic databases, not just a list of keywords.
4. Please use the PICOS or PECOS acronym for defining the research question and keywords for systematic review. Incorporate it into the methodology.
5. Specify explicitly inclusion and exclusion criteria.
6. What are “other study results”? Define.
7. Statistical analysis: report the way of presenting data.
Results
1. List reasons for the exclusion of 21 excluded articles into the Flow diagram
2. There is a problem with the number of excluded and finally included publications. If you had 50 and excluded 21, how can you include 30? Please check!
3. Lines: 154-157, some spaces missed.
4. You did not report within results: the total number of HBV patients, number of HBV patients according to genotype, according to region, etc.
5. Also, you did not report the study design of the included publications.
6. What about data regarding vaccination?
7. It looks like the sentence “Infected people can transmit the virus through their blood.” is the excess or you had an idea to incorporate this information within the interpretation of tests. Please reword.
8. There is no need to list results from different publications in results. It has to be reported collectively. Lines: 187-190. Please change this part with reports regarding the frequency distribution of HBV genotypes considering all included publications.
9. Why didn`t you try to perform a meta-analysis of seroprevalence of HBV? Please, try to do it and include this part of the statistical analysis in the methodology.
10. Line 203: change “in the course of our studies.” into “in the course of the included studies.”
11. Lines: 205-233. Report important results from the included studies collectively. How many studies evaluated the association? Which method do they use (chi-square test, logistic regression, etc.)? It is allowed to give a brief explanation of their results but not to list almost all of them. It is more appropriate for the discussion section.
12. You failed to extract data about HBV patients' age, gender, comorbidities (states of immunodeficiency, malignancies, transplantations, etc.) Please add them to the table and results.
Discussion
1. Please redefine the aim of this systematic review a bit. It does not deal with knowledge but with available epidemiological and public health data or information about HBV in DRC.
2. You did not appropriately emphasize the importance of your systematic review. It only gives a picture of the technically done work without an impact. The discussion section is the part where it can be done.
Comments on the Quality of English LanguageYour English requires a minor editing.
Round 2
Reviewer 1 Report
Comments and Suggestions for Authors Overall the manuscript looks much better.My comment is accepted with minor changes. My only recommendation is to have editor proofread language.Comments on the Quality of English Language My only recommendation is to have editor proofread language.
Author Response
Thanks for your comments.